# Formulation of an Efficiency Model Valid for High Vacuum Flat Plate Collectors

Eliana Gaudino [1,2], Antonio Caldarelli [1,2], Roberto Russo [2,*] and Marilena Musto [1,2]

1  Industrial Engineering Department, University of Napoli "Federico II", Piazzale Vincenzo Tecchio 80, 80125 Napoli, Italy; eliana.gaudino@na.isasi.cnr.it (E.G.); antonio.caldarelli@na.isasi.cnr.it (A.C.); marilena.musto@unina.it (M.M.)
2  Institute of Applied Sciences and Intelligent Systems, National Research Council of Italy, via Pietro Castellino 111, 80131 Napoli, Italy
*  Correspondence: roberto.russo@na.isasi.cnr.it

**Abstract:** High Vacuum Flat Plate Collectors (HVFPCs) are the only type of flat plate thermal collectors capable of producing thermal energy for middle-temperature applications (up to 200 °C). As the trend in research plans is to develop new Selective Solar Absorbers to extend the range of HVFPC application up to 250 °C, it is necessary to correctly evaluate the collector efficiency up to such temperatures to predict the energy production accurately. We propose an efficiency model for these collectors based on the selective absorber optical properties. The proposed efficiency model explicitly includes the radiative heat exchange with the ambient, which is the main source of thermal losses for evacuated collectors at high temperatures. It also decouples the radiative losses that depend on the optical properties of the absorber adopted from the other thermal losses due to HVFPC architecture. The model has been validated by applying it to MT-Power HVFPC manufactured by TVP-Solar. The dissipative losses other than thermal radiation were found to be mostly conductive with a linear coefficient k = 0.258 W/m$^2$K. The efficiency model has been also used to predict the energy production of HVFPCs equipped with new, optimized Selective Solar Absorbers developed in recent years. Considering the 2019 meteorological data in Cairo and an operating temperature of 250 °C, the annual energy production of an HVFPC equipped with an optimized absorber is estimated to be 638 kWh/m$^2$.

**Keywords:** solar energy; efficiency formula; thermal energy; solar thermal collectors; radiation losses; thermal losses; high vacuum flat plate collectors; HVFPC

## 1. Introduction

According to the International Energy Agency [1], industrial heat makes up two-thirds of industrial energy demand and almost one-fifth of global energy consumption. Since the majority of industrial heat is produced by fossil fuel combustion, it creates a substantial stream of $CO_2$ emitted every year [2]. Decarbonizing the heat production field requires a significant shift in generating industrial heat, especially in the high- and medium-temperature heating sectors [3]. Solar collectors' technology is constantly evolving, enabling the spread of their use to an increasing number of utilizations [4]. The trend of the annual global change in renewable heat consumption by source in 2022 [5] shows that a more significant portion of the 150 PJ of extra energy produced by solar thermal technology during the preceding year was used to satisfy the building sector energy demand and just a small amount of the industrial heat demand. Solar thermal energy systems can be classified into three categories regarding operating temperature: low-temperature systems (<150 °C), such as Conventional Flat Plate Collectors, Evacuated Tube Collectors, and Compound Parabolic Collectors; medium temperature systems (from 150 to 400 °C), such as Parabolic Trough Collectors, Linear Fresnel Collectors, and High Vacuum Flat Plate Collectors (HVFPCs); and high-temperature systems (>400 °C), such as

Large Parabolic Trough Collectors and Linear Fresnel Collectors [6]. Flat plate (FP) systems are the most fundamental and studied technology for solar-powered domestic hot water systems. The overall idea behind this technology is simple: they can collect diffuse and direct rays without solar tracking, being cheaper than concentrating collectors [7]. However, Conventional Flat Plates are mostly used in applications requiring temperatures lower than 100 °C, since they experience high thermal losses caused by conduction, convection, and radiation. The conduction heat losses occur from the sides and the back of the collector plate, the convection heat losses take place from the absorber plate to the glazing cover, and the radiation losses occur from the absorber plate to the envelop [8]. To minimize conduction loss, materials with low thermal conductivity should be used, whereas a drastic reduction in convective losses to a negligible value can be obtained by evacuating the space around the absorber plate [9]. A high vacuum-insulated collector means that the pressure inside the collector is maintained below $10^{-4}$ mbar [10]. High vacuum insulation technology allowed for the development of highly efficient solar collectors in the solar thermal sector, with High Vacuum Flat Plate Collectors like SRB [11] and TVP-Solar MT-Power collectors [12], or even hybrid photovoltaic thermal systems like virtu collector manufactured by Naked Energy.

As for their features, HVFPCs, compared with conventional thermal collectors, exhibit a reduction in the heat loss coefficient, resulting in a noticeable increase in efficiency [13,14] and executable thermal performance in industrial applications. They also achieve better solar conversion efficiency than the concentrating collectors, particularly in the regions with a high proportion of diffuse solar irradiance [15,16]. High vacuum insulation in a panel with a flat structure can combine the high fill factor and the ease of building integration with low heat losses. Consequently, evacuated collectors can cover a larger range of applications that require thermal heat. Despite the noticeable advantages, few manufacturers produce an industrial version of flat collectors under high vacuum because dealing with vacuum requires great attention to collector architecture design. In an HVFPC, the flat absorber is contained within an evacuated enclosure with a glass cover on top supported against atmospheric pressure loading. Even if high-vacuum insulation suppresses convection, the radiative heat exchange is still present and represents the main heat exchange mechanism in a vacuum environment. It becomes more significant with the rise in the operating collector temperature. The importance of the radiative behavior of an HVFPC makes the adoption of a Selective Solar Absorber (SSA) necessary to guarantee high absorptance in the solar irradiation spectrum (λ = 0.3–2.5 μm) and low emissivity in the infrared region (IR) (λ > 2.5 μm), factors that reduce the drop in the absorber efficiency at high temperatures [17,18]. The low thermal emittance of the selective absorber also reduces the radiative losses, resulting in a very high thermal efficiency, which is the most significant parameter to express the performance of a thermal solar collector [19]. Its most general expression is the ratio between the useful thermal power $P_U$ transferred to the heat transfer fluid (HTF) and the incidental solar irradiation G on the collector aperture surface $A_c$ [20]:

$$\eta_{th} = \frac{P_U}{G \times A_c} \qquad (1)$$

At present, the most common expression (Equation (2)) to describe the flat plate collector efficiency approximates thermal losses to a second-order dependence from the difference between the average of the working fluid temperatures at the inlet and the outlet of the collector ($T_m$) and the ambient temperature ($T_{amb}$) [21]. This expression is also the most used in Solar Keymark certification, and, in this manuscript, will be indicated as standard efficiency and expressed as Equation (2):

$$\eta_{th\_st} = \eta_0 IAM_\theta - \left[ \frac{c_1(T_m - T_{amb}) + c_2(T_m - T_{amb})^2}{G} \right] \qquad (2)$$

Here, $\eta_0$ is the zero-loss efficiency (when the fluid temperature is identical to the ambient temperature $T_{amb}$). The factors $c_1$ and $c_2$ are the first- and second-order heat loss coefficients, respectively. The incidence angle modifier ($IAM_\theta$) function describes the optical efficiency for a certain radiation incidence angle $\theta$ normalized by optical efficiency evaluated at perpendicular irradiation conditions [22].

The standard efficiency coefficients ($\eta_0$, $IAM_\theta$, $c_1$, $c_2$) are provided by the Solar Keymark certification body that obtains them through a test whose methodology is prescribed by EN 12975 [23]. All relevant collector properties and characteristics are reported in the Solar Keymark database [24]. The Solar Keymark certification also indicates the stagnation temperature value of the collector (equilibrium temperature where $\eta = 0$). Efficiency values for operating temperatures higher than the maximum tested temperature cannot be evaluated with standard efficiency curve extrapolation since the radiative losses are not taken explicitly into account by Equation (2).

Figure 1 shows the MT-Power efficiency curve certified by the Solar Keymark up to $T_m = 200\ °C$ (solid line) and its extrapolation obtained with formula (2) until the zero-efficiency point (dashed line). Additionally, the figure includes a black arrow indicating the specific point that is the certified stagnation temperature of the MT-Power ($T_m = 302\ °C$).

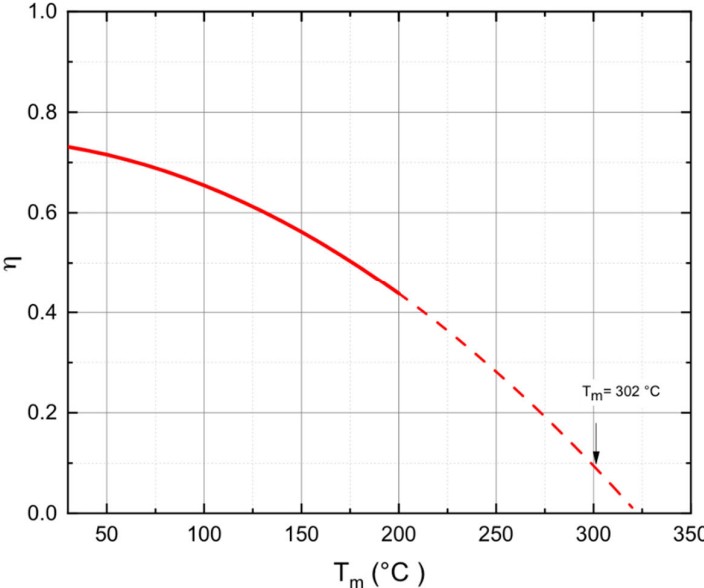

**Figure 1.** Standard thermal efficiency curve of the TVP-Solar HVFPC MT-Power and its mathematical extrapolation (dashed line). The certified collector stagnation temperature ($T_m = 302\ °C$) is also indicated in the figure with the black arrow.

The curve represented in Figure 1 is described by the following Equation (3):

$$\eta th\_st\_MTPower = 0.732 - \left[ \frac{0.5(Tm - 20) + 0.006(Tm - 20)^2}{950} \right] \qquad (3)$$

Figure 1 shows that, in correspondence with the certified stagnation temperature, the mathematical extrapolation of the standard curve overestimates the thermal efficiency by a value of 0.1, indicating that the extrapolation overestimates the efficiency and cannot be used to predict the collector behavior at higher temperatures. Currently, HVFPCs are used to produce heat up to 200 °C, but the improvements in this technology, like the development of new optimized SSA coatings [24,25] and the use of a glass cover [26,27], will allow the ability to cover applications up to 250 °C [28] in short order. A reliable method for evaluating the annual energy production of an HVFPC equipped with a new SSA would allow for estimations regarding the potential economic advantages of adopting

the new absorber. Such a method is missing in the literature, and it would be essential to provide insights into how much to invest in adopting this innovation.

This paper fills this gap and presents a novel efficiency model for the performance characterization of HVFPCs; it overcomes the limits of the standard efficiency formula, allowing the efficiency calculation up to the stagnation temperature.

The proposed formula decouples the losses that depend on collector architecture from the optical and radiative losses of the SSA. It will allow us to predict the HVFPC performances at temperatures higher than the certified temperature and to accurately estimate the efficiency of HVFPCs equipped with the new SSAs with different properties.

## 2. Materials and Methods

This paragraph describes the efficiency model developed for the energetic characterization of High Vacuum Flat Plate Collectors (HVFPCs). The proposed model is applicable to various HVFPCs and is derived from an energy balance analysis of the thermal system known as the "High Vacuum Flat Plate Solar Collector".

### 2.1. Efficiency Model for HVFPCs

A thermal solar collector can be defined as a device that converts solar power into useful heat, increasing the internal energy, i.e., the temperature of an HTF that will be used for the end-user application. The energy transfer from solar radiation to the HTF will occur through different heat exchange mechanisms, as shown in Figure 2.

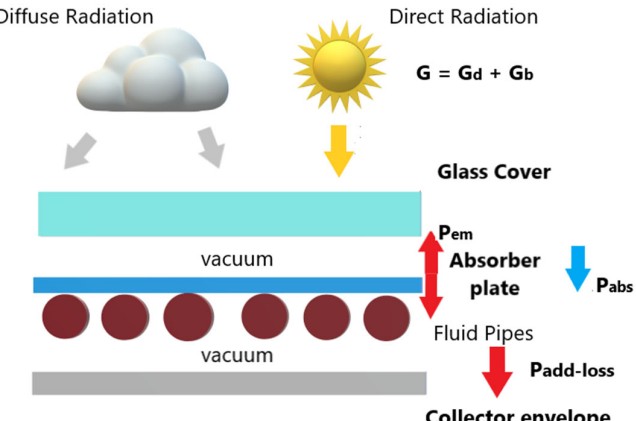

**Figure 2.** Simplified scheme of basic components of an HVFPC and thermal fluxes at interfaces.

The main losses in HVFPCs are radiative losses, which depend on the absorber radiative properties (absorptance and emittance) and the glass cover transmittance, and additional thermal losses, mostly conductive, due to the thermal contact with the supporting structure. Therefore, $P_u$ can be expressed as

$$P_u = [P_{abs} - (P_{em} + P_{add-loss})] \tag{4}$$

where $P_{abs}$ represents the amount of power transmitted by the glass, absorbed by the SSA plate, and converted into HTF internal energy increment. It can be expressed as the product of the collector zero-loss efficiency $\eta_0$ and the incidence angle modifier $IAM_\theta$:

$$P_{abs} = \eta_0 \times IAM_\theta \tag{5}$$

$P_{em}$ is the absorber emitted power due to the radiative heat exchange between the absorber and the surrounding elements:

$$P_{em} = A_{abs} \times \left\{ \varepsilon_{eabs}(T_m)\sigma \left[ (T_m + 273.15)^4 - (T_g + 273.15)^4 \right] \right.$$
$$\left. + \varepsilon_{esub}(T)\sigma \left[ (T_m + 273.15)^4 - (T_v + 273.15)^4 \right] \right\} \tag{6}$$

In Equation (6), $T_m$ is the average absorber temperature, which will be considered equal to the average HTF temperature. Equation (6) considers the effective thermal emittance of the two sides. The first ($\varepsilon_{eabs}$) refers to the absorbing side facing the glass at $T_g$, and the second ($\varepsilon_{esub}$) refers to the substrate side facing the stainless-steel vessel at temperature $T_v$.

For the surfaces of the components of a flat collector, the model of infinite parallel plates [29] can be used to calculate the effective emittances $\varepsilon_e$ that govern the radiative exchange between them, so $\varepsilon_{eabs}$ and $\varepsilon_{esub}$ can be written as

$$\varepsilon_{eabs} = \frac{1}{\frac{1}{\varepsilon_{abs}} + \frac{1}{\varepsilon_{glass}} - 1} \tag{7}$$

$$\varepsilon_{esub} = \frac{1}{\frac{1}{\varepsilon_{sub}} + \frac{1}{\varepsilon_{vessel}} - 1} \tag{8}$$

Experimental measurements showed that the difference between $T_g$, $T_v$, and $T_{amb}$ never exceeds 10 °C, impacting less than 1% on the radiated power. For this reason, we can safely assume $T_g = T_v = T_{amb}$, and Equation (6) becomes

$$Pem = \varepsilon_e(Tm)\sigma Aabs \left[ (T_m + 273.15)^4 - (T_{amb} + 273.15)^4 \right] \tag{9}$$

where $\varepsilon_e$ is the sum of $\varepsilon_{eabs}$ and $\varepsilon_{esub}$ and is the total effective emittance of the absorber. The term $P_{add-loss}$ of Equation (4) included all the HVFPC losses due to the panel architecture.

$$P_{add-loss} = k(T_m - T_{amb}) \tag{10}$$

where $k$ is the conductive heat losses coefficient while the exponent $z$ determines the type of the additional heat losses, and it is expected to be equal to one (mostly conductive heat losses).

Explicating all the terms, Equation (1) becomes Equation (11):

$$\eta_{th,new} = \eta_0 \times IAM_\theta - \left\{ \frac{\varepsilon_e(T_m) \times \sigma \left[ (T_m + 273.15)^4 - (T_{amb} + 273.15)^4 \right]}{G} \frac{A_{abs}}{A_c} + \frac{k(T_m - T_{amb})^z}{G} \right\} \tag{11}$$

Equation (11) is the expression of the proposed efficiency model valid for HVFPCs. The novelty of the Equation (11) is that it decouples the absorber radiative losses from the losses due to the HVFPC architecture (with the variation of parameter $k$, specific for a given HVFPC design). A particular mention ought to be made to its adaptability to different absorber coatings when considering the panel architecture unchanged: by changing the absorber radiative properties in Equation (11), it is possible to quantify the change in the collector thermal efficiency and correctly estimate the energy production.

## 2.2. Application of Proposed HVFPCs Efficiency Model to MT-Power TVP-Solar Collector

In this work, the proposed efficiency model devised for HVFPCs will be validated by its application to the MT-Power collector produced by TVP-Solar company. The enclosure follows a 'tray' style design, employing a stainless-steel tray with a single cover glass on the front, schematically shown in Figure 3a. From top to bottom, the components include the anodized aluminum frame, highly transparent tempered glass, a high vacuum glass–metal

seal, a selective coated heat absorber, heat transfer fluid pipes, atmospheric pressure supports (that pass through holes in the absorber), and a back metal sheet. Approximately 3% of the absorber area is occupied by the holes, leaving 97% of the 'absorber area' to absorb heat. In Figure 3b, the fitted thermal emittance curve of the selective coated commercial heat absorber is depicted.

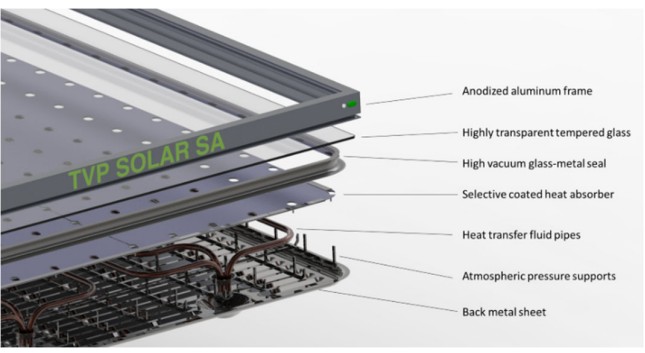

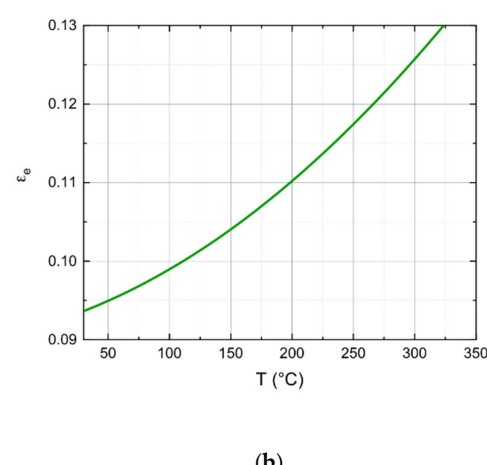

(**a**)     (**b**)

**Figure 3.** (**a**) Schematic of TVP-Solar HVFPC MT-Power Structure; (**b**) thermal emittance as function of temperature for the commercial SSA mounted on MT-Power.

The curve represents the variation in thermal emittance with respect to the absorber temperature, providing important insights into the performance characteristics of the heat absorber. The absorber emittance curve was experimentally obtained by averaging the thermal emittance measured with a calorimetric approach [30,31]. The MT-Power operates at a maximum temperature of 200 °C and employs a commercial absorber with a multi-layer structure consisting of an aluminum substrate coated with an absorption layer and an anti-reflection layer.

When deriving the thermal efficiency formula of MT-Power, the optical properties of the SSA, such as absorptance $\alpha$ and effective emittance $\varepsilon_e(T)$, were obtained through calorimetric measurements following the procedure described in [31,32].

The spectrally averaged absorptivity ($\alpha$) remains constant at 0.95 regardless of the absorber average temperature ($T_m$). However, the $\varepsilon_e(T_m)$ exhibits a quadradic dependence on absorber plate temperature, as shown in Figure 3b.

Given the known SSA optical properties, the unknown terms in Equation (11) are the additional losses coefficient ($k$) and the exponent ($z$). To determine the values of $k$ and $z$, Equation (11) can be fitted to the certified efficiency data obtained using the standard efficiency formula of the collector.

The standard efficiency curve is certified in the range between the minimum and the maximum tested temperature.

The proposed efficiency model equation (Equation (11)) should accurately replicate the standard formula (Equation (2)) until $T_m = 200$ °C.

## 3. Results and Discussion

In the following section, we will present the efficiencies obtained using the proposed model and will compare them to the standard and/or optical efficiencies. Furthermore, we will calculate the energy production for the different efficiencies.

### 3.1. MT-Power HVFPC Efficiency

Through the fitting procedure, it was determined that the exponent ($z$) of the additional losses function that reproduces the certified efficiency is 0.93 (+/−0.08) with a $\chi^2$ value of $2.8 \times 10^{-5}$. This value confirms that dissipative effects other than radiation losses have

a linear dependence on $(T_m - T_{amb})$, suggesting that they are due to thermal conduction. Since the fit result is compatible with the exponent 1 of the conductive loss, we fix the value of z at 1 to compute the coefficient of additional losses, k. The best fit was obtained for a k value equal to 0.258, and it returned a $\chi^2$ value of $2.8 \times 10^{-5}$, and $R^2 = 0.999$, which was identical to the previous ones. The low value of the additional losses' coefficient is not surprising due to the presence of a high vacuum in the panel and due to the advanced architecture of MT-Power, specially designed to minimize conductive losses.

The efficiency formula of the MT-power collector obtained from the best fit of the Solar Keymark data is reported here:

$$\eta_{th} = \eta_0 \times IAM_\theta - \left[ \frac{\varepsilon_e(T_m) \times \sigma \left( T_m^4 - T_{amb}^4 \right)}{G} \times \frac{A_{abs}}{A_c} + \frac{0.258(T_m - T_{amb})}{G} \right] \quad (12)$$

Figure 4 shows three curves: the standard efficiency curve (red line) extended up to the stagnation temperature (red dashed line), the optical efficiency curve (yellow line) representing radiative losses only, and the curve generated by Equation (11) (blue line with dots). All curves are plotted for normal irradiance at an angle of incidence of θ = 0. The x-axis of Figure 4 represents the average temperature $T_m$ because the optical efficiency curve and Equation (12) depend on the fourth power of the ambient temperature $T_{amb}$ and the absorber temperature $T_m$ separately. To plot a single efficiency curve, $T_{amb}$ must be fixed. In this case, Figure 4 uses the value at which the standard test was performed (indoor measurements), i.e., $T_{amb} = 20\,^\circ$C.

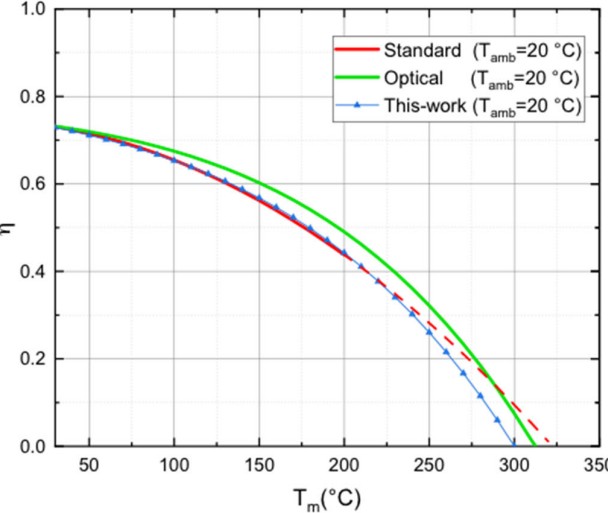

**Figure 4.** TVP-Solar MT-Power HVFPC efficiency curves: certified (red line) and its extrapolation up to zero efficiency (red dashed line), optical (green), and the curve obtained in this work (blue line with triangles).

It is important to highlight that, unlike the standard curve extrapolation (Figure 1), Equation (12) reaches zero efficiency at $T_m = 302\,^\circ$C, which corresponds to the collector's stagnation temperature stated in the Solar Keymark certification.

Our proposed efficiency formula, Equation (12), corrects the absence of the quartic temperature dependence in the standard thermal efficiency formula. Using the standard formula, the power available at 250 °C amounts to 525 W, while employing our proposed efficiency formula yields 464 W, resulting in a difference of 61 W. This discrepancy increases notably with higher operating temperatures $T_m$. Furthermore, the same fitting procedure for Equation (11) can be applied to HVFPCs with different architectures, given that we know the absorber optical properties and the standard efficiency coefficients from the Solar Keymark certification. The fitting process will provide the additional losses coefficient 'k' and the efficiency equation that can be used to extrapolate the collector performance

up to stagnation. The stagnation temperature in the certification is vital information to obtain the new efficiency equation. The fact that the obtained equation passes through this specific point serves as a significant validation of the efficiency extrapolation outside the temperature range explored by the certification.

*3.2. Application of Proposed HVFPCs Efficiency Model to HVFPC Equipped with New Optimized Solar Absorbers*

The proposed model allows us to predict the efficiency of an HVFPC equipped with a new SSA with different optical properties. This feature is crucial because absorber properties can change over time due to aging. Additionally, there are ongoing developments of new types of Selective Solar Absorbers that enable efficient operation at temperatures higher than 200 °C.

In this section, we apply the newly developed efficiency model to an MT-Power HVFPC architecture equipped with a multi-layered SSA optimized for operating temperatures of 200 °C and 300 °C [28]. To achieve maximum efficiency at the selected operating temperatures, the thickness of each layer was determined using a genetic algorithm [28]. The optimized absorbers exhibit optical properties suitable for HVFPCs, ensuring a significantly low emittance at high temperature. In Figure 5a, the fitted thermal emittance of the commercial absorber mounted on TVP-Solar HVFPCs is represented by the black dashed line. Additionally, the thermal emittances of selectively coated solar absorbers (SSAs) optimized to operate at different temperatures are shown. The green dashed line corresponds to optimization for a 200 °C operating temperature, while the red dashed line represents optimization for a 300 °C operating temperature.

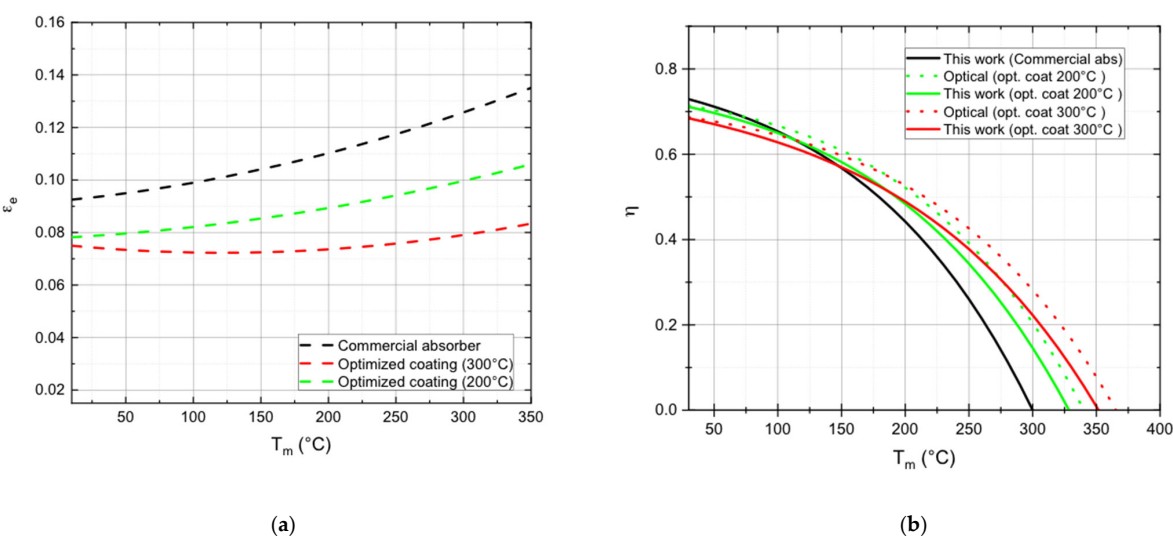

(a)                                                                      (b)

**Figure 5.** (**a**) Thermal emittance as function of the operating temperature for the following SSA: commercial SSA mounted on TVP-solar HVFPCs (black dashed line), and SSAs optimized to work at 200 °C (green dashed line) and 300 °C (red dashed line); (**b**) thermal efficiency curves versus operating temperature for a TVP-solar HVFPC equipped with commercial (black line) and SSAs optimized to work at 200 °C (green line) and 300 °C (red line) obtained with the proposed efficiency model. For the optimized SSAs, the optical efficiency curves are also reported (dot lines).

The optimization led to a significant reduction in thermal emittance, which is the primary source of heat loss for HVFPCs at elevated temperatures [25]. However, this reduction came with a slight trade-off in absorptance, with values of $\alpha_{comm} = 0.95$, $\alpha_{opt200} = 0.925$, and $\alpha_{opt300} = 0.890$.

In Figure 5b, the curves labelled "this work" depict the MT-Power with a commercial absorber (black continuous line) and the HVFPCs with the same structural characteristics as the MT-Power but equipped with SSAs optimized for 200 °C (green continuous line) and 300 °C (red continuous line). Due to the different value of $\alpha$, HVFPCs equipped with the optimized SSAs exhibit zero-loss efficiencies ($\eta_0$) different from those of the HVFPCs

equipped with commercial absorbers. The efficiency of the optimized SSA is consistently higher than that of the commercial absorber at temperature higher than 120 °C, since the latter was not optimized to work in high vacuum. The optical efficiencies of the optimized cases [28] are also reported in Figure 5b as dashed lines to underscore the importance of accounting for conductive losses, especially at elevated temperatures.

### 3.3. HVFPCs Annual Energy Producibility Calculation

The thermal efficiency of a solar thermal collector is the crucial parameter for energy predictions and estimating the collector's production under specific climatic conditions. Failing to account for radiative losses in the efficiency equation can lead to inaccuracies in estimating the collector's performance, especially at high operating temperatures.

Figure 6 presents the monthly energy production of the MT-Power at $T_m = 250$ °C, using 2019 hourly irradiation and ambient temperature data from a specific location (Cairo). The calculation considers the energy converted by a collector that is oriented to the south and tilted at 35°. In Figure 6a, three distinct efficiency formulas were employed: optical (illustrated by green line squares), this-work (represented by blue line squares), and standard (depicted by red line squares). It is apparent that relying on the standard efficiency formula results in an overestimation of energy production.

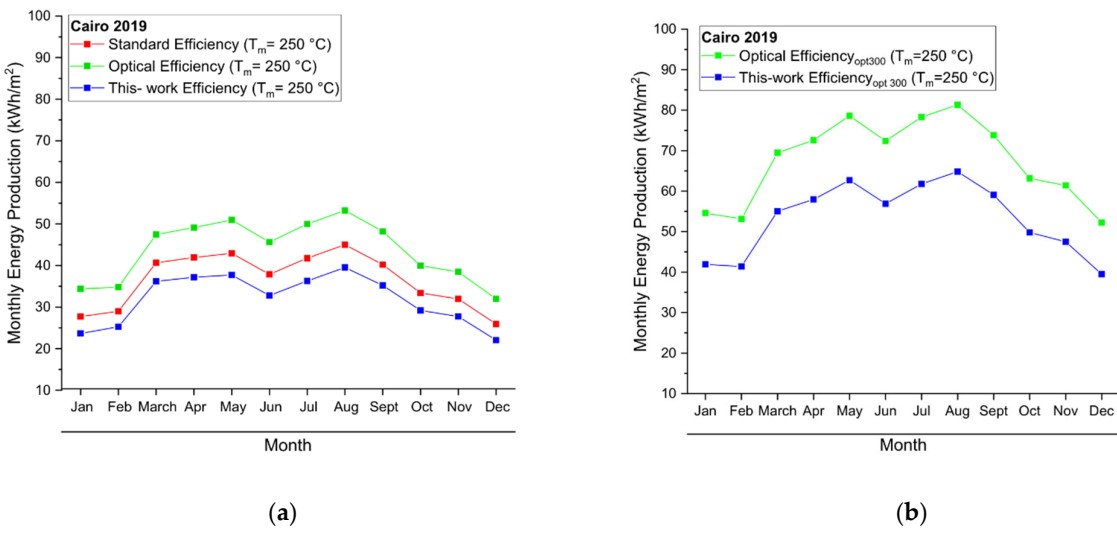

(**a**)                                                                 (**b**)

**Figure 6.** Monthly energy production (**a**) for MT-power collector computed using optical (green), standard (red), and this-work (blue) efficiencies; (**b**) for HVFPC with SSA optimized at $T_m = 300$ °C, computed using optical (green) and this-work (blue) efficiencies.

To be specific, for the year 2019, an annual energy production of 438 kWh/m$^2$ would be projected, while in contrast, using the this-work efficiency formula, a more conservative estimate of 383 kWh/m$^2$ would be indicated, leading to a difference of roughly 13%.

Figure 6b underscores the importance of accounting for additional conductive losses in the efficiency calculations of HVFPCs. This figure highlights the variance in the monthly energy production of an HVFPC fitted with a new absorber (optimized at 300 °C) and set at an operating temperature of 250 °C.

The comparisons are drawn between results obtained using the optical efficiency (green dots and line) and those derived from the proposed this-work efficiency model (blue dots and line). The estimated annual energy production, as per the optical efficiency formula, stands at 811 kWh/m$^2$. In contrast, the this-work efficiency formula suggests a value of 638 kWh/m$^2$.

### 4. Conclusions

This manuscript introduces a thermal efficiency model tailored to predict the performance of High Vacuum Flat Plate Collectors (HVFPCs). Unlike traditional models, the

proposed framework gives explicit attention to radiative losses, the predominant energy loss mechanism in high vacuum collectors. These losses are inherently linked to the optical properties of the absorber that can be measured independently. The efficiency model equation (Equation (11)) includes the absorber optical properties and an additional heat conductive loss coefficient k, which can be determined using the standard efficiency coefficients ($\eta_0$, $c_1$, and $c_2$) specific to the HVFPC being analyzed. To validate the proposed efficiency model, it was applied to the TVP-Solar MT-Power HVFPC and its precision was confirmed when the zero-efficiency point for the MT-Power matched the collector's certified stagnation temperature (302 °C). Therefore, it is evident that the presented thermal efficiency model for HVFPCs is capable of computing thermal efficiency across the full spectrum of operating temperatures, extending to the collector's stagnation temperature, provided the optical properties of the absorber are known.

Moreover, the model was employed to predict the efficiency of HVFPCs potentially equipped with different Selective Solar Absorbers (SSAs) optimized to meet heat demands at 200 °C and 300 °C.

This predictive capability is not present in the efficiency formula that does not include the radiative term. This predictive aptitude paves the way for more accurate estimations of the annual energy output of these optimized HVFPCs, as illustrated in Figure 6a,b.

In conclusion, the proposed efficiency model for HVFPCs is flexible and can provide valuable insights for designing and optimizing SSAs for HVFP collectors, taking into account their annual energy production.

**Author Contributions:** Conceptualization, methodology, data curation and writing-original draft preparation, E.G.; data curation, formal analysis, and methodology, A.C.; conceptualization, supervision and writing–review and editing, R.R.; visualization supervision and writing-review and editing, M.M. All authors have read and agreed to the published version of the manuscript.

**Funding:** The Ph.D. grant of E.G. is funded by the CNR-Confindustria "Dottorati di Ricerca Industriali" program XXXVI ciclo. This study was partially funded by the Eurostar Program powered by EUREKA and the European Community (Project ESSTEAM reference E! 115642 CUP B69J21036070005).

**Data Availability Statement:** Hourly 2019 Cairo solar irradiation and ambient temperature data are available at Photovoltaic Geographical Information System (PVGIS) tool at https://re.jrc.ec.europa.eu/pvg_tools/it/#HR (accessed on 20 October 2023). MT-Power efficiency coefficients and standard performance test information are available in Solar Keymark database at https://solarkeymark.eu/database (accessed on 20 October 2023).

**Conflicts of Interest:** The authors declare no conflict of interest.

## Nomenclature

| | |
|---|---|
| $A_{abs}$ | Absorber Surface (m$^2$) |
| $A_c$ | Collector Aperture Surface (m$^2$) |
| $c_1$ | First-Order Heat Loss Coefficient (W/m$^2$ K) |
| $c_2$ | Second-Order Heat Loss Coefficient (W/m$^2$ K$^2$) |
| $c_p$ | Specific Heat at Constant Pressure (kJ/kg K) |
| $G$ | Solar Irradiation (W/m$^2$) |
| $m$ | Mass Flow Rate (kg/s) |
| $P_{abs}$ | Absorbed Power (W) |
| $P_{add\text{-}loss}$ | Lost Power Due to Conductive Losses (W) |
| $P_{em}$ | Emitted Power (W) |
| $P_U$ | Useful Power (W) |
| $T_m$ | Average Temperature (°C) |
| $T_{amb}$ | Ambient Temperature (°C) |
| $T_g$ | Glass Temperature (°C) |
| $T_v$ | Vessel Temperature (°C) |

*Abbreviations*

| | |
|---|---|
| abs | Absorber |
| comm | Commercial |
| DHW | Domestic Hot Water |
| HTF | Heat Transfer Fluid |
| HVFPC | High Vacuum Flat Plate Collector |
| IAM | Incidence Angle Modifier |
| SSA | Selective Solar Absorber |
| sub | Substrate |
| th | Thermal |
| IR | InfraRed |
| Symbols | |
| $\alpha$ | Spectrally Averaged Absorptivity |
| $\varepsilon$ | Spectrally Averaged Emissivity |
| $\eta$ | Efficiency |
| $\eta_0$ | Zero-Loss Efficiency |
| $\sigma$ | Stefan–Boltzmann Constant (W/m$^2$K$^4$) |
| $\lambda$ | Wavelength |
| k | Conductive Heat Losses Coefficient (W/m$^2$K) |

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
