# Peer review of "Formulation of an Efficiency Model Valid for High Vacuum Flat Plate Collectors"

_energies, doi:10.3390/en16227650_

Round 1

Reviewer 1 Report

Comments and Suggestions for Authors

REVIEW (2023-10-26):

REVIEWER'S ASSUMPTIONS: The article deals with the functionality of the flat plate vacuum collector system, and in detail about the inaccuracies of the functions it performs. Efficiency is the susceptibility to useful action, the useful load to heating.  The efficiencies of high-vacuum flat collectors, solved on mathematical models, require a systemic (mathematical) approach. Models should include the overall integrated efficiency of the system, and/or the component efficiencies of: the process system P, which is responsible for the phenomenal functioning, i.e. the achievement of the set goal by the system (useful work approximately equal to the input energy); control system C, responsible for high-efficiency power supply, control and coordination of process, information and logistics activities; information system I, self-organizing the processing and distribution of information streams inside and outside - in line with the needs and objectives of the other subsystems; L logistics system, guaranteeing the reliable maintenance and supply of systems (including itself), for the reliable achievement of numerous goals optimally configured technical conditions, vacuum flat collector.

ADEQUATE KNOWLEDGE: The article presents adequate, multi-faceted, partially coordinated and susceptible to optimal technical conditions, a fragmentary study of the issues of the process system (P) and logistics system (L) based on 31 bibliographic items. It was assumed that in order to minimize conduction losses, materials with low thermal conductivity should be used, while a drastic reduction of convection losses to a negligible value can be achieved by emptying the space around the absorber plate. The most important model is the ratio of the useful heat power Pu transferred to the heat transfer fluid (HTF) to the power (energy) of the incident solar radiation G to the active surface of the collector Ac. The attractiveness of the model (10) lies in the separation of absorber radiation losses from losses resulting from geometric, material, dynamic and partly environmental design features of the HVFPC (changes in parameter k, specific to the HVFPC structure). Hence the susceptibility to different absorber coatings, for the constant characteristics of the panel. By changing the radiative properties of the absorber in Equation (10), it is possible to model the quantitative change in the thermal efficiency of the collector and correctly estimate the energy production.

RESEARCH SKILLS AND INSTRUMENTS: Many papers present energy support (including energy generation from RES, charging and discharging energy storage), based on determined, probabilistic and stochastic mathematical models, as well as models adapted to determine and analyze reliability, availability and safety under the constraints of models and computational tools as well as existing data on energy generation and consumption. The proposed efficiency model, Equation (11), corrects for the lack of dependence on the temperature of the quart in the standard formula for thermal efficiency. The use of the proposed model corrects the heat output performance, whereby the correction is also significant at higher operating temperatures Tm, so it can be applied to HVFPC of different designs, taking into account the known optical properties.

PURPOSEFULNESS AND CREATIVE ATTITUDE: The example of deliberate support of the upper limit of collector efficiency, the so-called optical efficiency, is of great importance for the determination of data and characteristics from laboratory tests in accordance with the EN 12975 standard. The parameters included in the Solar Keymark certificate are: optical efficiency (η0), linear heat loss coefficient (c1), nonlinear heat loss coefficient (c2), solar collector stagnation temperature (tstg). The example of MT-Power highlights the inclusion of additional conduction losses in the HVFPC efficiency calculations, the variability of the monthly energy production of the HVFPC equipped with the new absorber (optimized for thermal conditions of 300°C), the comparisons between the results obtained using optical efficiency and the results obtained from the proposed efficiency model. If the thermal insulation of the collector was so perfect that the heat loss would be zero, then the efficiency diagram would be horizontal.  That is, the efficiency regardless of the temperature of the absorber and the environment would always be the same and equal to the optical efficiency.  The higher the efficiency is, the lower the temperature of the absorber (and at the same time the temperature difference between the absorber and the environment). The most beneficial effects are achieved by solar collectors in low-temperature systems.

CONCLUSION: Critical remarks: I do not have any critical remarks, I did not have a clear distinction between the functionality of the system (especially the inaccuracy of the collector function) and the efficiency of the collector (the relation of useful work, susceptibility to useful action (heating of objects), e.g. the working load to the energy put in by the sun. 

I rate the work positively!

Author Response

we thank you the reviewer for the carefully reading of the manuscript and for having appreciated our work.

Reviewer 2 Report

Comments and Suggestions for Authors

The article titled "Formulation of an efficiency model valid for High Vacuum Flat Plate Collectors" has been revised.

Issuing the following comments in the spirit of improving the manuscript

A lot of information is needed in the methodological section:

1.- The device to be evaluated, dimensions, is not adequately described. Etc.

2.- It is not described under what conditions this device was evaluated "although it is said that it is based on the standard"

It is necessary to clarify, mass or volumetric flows. radiation... in the results it is expected to see a graph that shows the climatological conditions of evaluation

3.- The location and some other experimental characteristics are not described (that is, under laboratory conditions or under real conditions)

4.- The place of experimentation is not specified

5.- No equipment used to measure operational parameters is described.

6.- The technical characteristics of the equipment used are not specifically described.

For equation 8, then it will only be valid for temperatures Tg, Tv and Tamb that do not exceed 10°C difference between them? What happens if it is greater, what error is expected to have?

Figure 3 a should be in the materials and methods section

Why use the standard method for determining heat transfer coefficients?

Methodological descriptions are observed in the results section, please review the section to promptly rearrange the information provided. Example last paragraph on page 12

Figure 6 and 7 remove the grid

Conclusions: other parameters can be considered to have a better characteristic curve for these collectors, since only considering the optical efficiency suggests assuming operating states.

Reviewer 3 Report

Comments and Suggestions for Authors

1- The abstract should be rewritten again and emphasize the most important findings from the study.

2- Correct the keywords

3- The font size must be unified in the same paragraph; review the paper.

4- Equation 2 is wrong. correct it.

5- Figure 1 caption is not acceptable. it includes an explanation. How is this?

6- All equations must be revised, with many errors in them. 

7- All captions are not accepted. It must be rewritten again according to the standard for scientific papers.

Comments on the Quality of English Language

The English needs to be reviewed well. 

Reviewer 4 Report

Comments and Suggestions for Authors

The article presents modified idea of a thermal balance flat plate solar collectors. Despite traditional Hottel-Whillier-Bliss equation authors focuses on radiative losses, the predominant energy loss mechanism in high vacuum collectors (HVFPC). To validate the proposed model, it was applied to the TVP-Solar solar collector which was tested according to demands of EN 12975. Proposed model confirmed its ability to calculate efficiency at high temperature range and reached the collector's certified stagnation temperature (302°C). The model is simply, easy to use and could be recommended to future investigation of selective layers of absorbers of flat solar collectors. The article can be accepted in present form.

Author Response

Thank you for the carefully reading of our manuscript and for having appreciated our work.

Round 2

Reviewer 3 Report

Comments and Suggestions for Authors

Comments on

Formulation of an efficiency model valid for High Vacuum Flat

Plate Collectors

The paper needs a revision; the following points could help the authors.

1-   Equation 2 is not right; what are the units of c1 and c2? If they have the same units, the equation must be corrected.

2-     Figure 6, the caption is not acceptable. Reduce it, the legend includes the required data, and you can add the details in the text not in the caption.

3-   Enhance the resolution of the figures. It is very bad and must be enhanced. Therefore, improve the figure in the manuscript.

Author Response

Thank you for carefully reading the manuscript.
Here are the answers to your comments:
1) the units of C1 and C2 have been included in the nomenclature in the new revised version of the manuscript (now the equation is dimensionally correct)
2) the caption of Figure 6 has been adjusted
3) all figures have be improved and included in the file in high resolution
thanks again